# Controlled Synthesis of Up-Conversion NaYF_4_:Yb,Tm Nanoparticles for Drug Release under Near IR-Light Therapy

**DOI:** 10.3390/biomedicines9121953

**Published:** 2021-12-20

**Authors:** Edelweiss Moyano Rodríguez, Miguel Gomez-Mendoza, Raúl Pérez-Ruiz, Beatriz Peñín, Diego Sampedro, Antonio Caamaño, Víctor A. de la Peña O’Shea

**Affiliations:** 1Signal Theory and Communications and Telematic System and Computing, Rey Juan Carlos University, Cam/del Molino 5, 28942 Madrid, Spain; edelweiss.moyano@urjc.es; 2Photoactivated Processes Unit, IMDEA Energy Institute, C/Ramón de la Sagra, 3, 28935 Madrid, Spain; miguel.gomez@imdea.org (M.G.-M.); raupreru@qim.upv.es (R.P.-R.); 3Departamento de Química, Universitat Politècnica de València, Camino de Vera S/N, 46022 Valencia, Spain; 4Departamento de Química, Centro de Investigación en Síntesis Química (CISQ), Universidad de La Rioja, Madre de Dios 53, 26006 Logroño, Spain; beatriz.penin@unirioja.es (B.P.); diego.sampedro@unirioja.es (D.S.)

**Keywords:** Up-Conversion, drug-delivery, Near IR-Light Therapy

## Abstract

Up-Conversion materials have received great attention in drug delivery applications in recent years. A specifically emerging field includes the development of strategies focusing on photon processes that promote the development of novel platforms for the efficient transport and the controlled release of drug molecules in the harsh microenvironment. Here, modified reaction time, thermal treatment, and pH conditions were controlled in the synthesis of NaYF_4_:Yb,Tm up-converted (UC) material to improve its photoluminescence properties. The best blue-emission performance was achieved for the UC3 sample prepared through 24 h-synthesis without thermal treatment at a pH of 5, which promotes the presence of the β-phase and smaller particle size. NaYF_4_:Yb,Tm has resulted in a highly efficient blue emitter material for light-driven drug release under near-IR wavelength. Thus, NaYF_4_:Yb,Tm up-converted material promotes the N-O bond cleavage of the oxime ester of Ciprofloxacin (prodrug) as a highly efficient photosensitized drug delivery process. HPLC chromatography and transient absorption spectroscopy measurements were performed to evaluate the drug release conversion rate. UC3 has resulted in a very stable and easily recovered material that can be used in several reaction cycles. This straightforward methodology can be extended to other drugs containing photoactive chromophores and is present as an alternative for drug release systems.

## 1. Introduction

The release of drugs through the blood to a target is a field widely investigated in recent decades [1,2]. The use of drug delivery systems by means of liposomes [3,4], mixed micelles [5,6,7], niosomes [8], micelles [9], bile salts aggregates [10,11,12], nanoparticles [13,14], nanocapsules, gold nanoparticles, microspheres [15], microcapsules, nanobubbles [16], microbubbles, and dendrimers [17,18] are being investigated for diagnosis and therapy [19,20]. Carriers based on micelles (liposomes, mixed micelles, and niosomes, among others) are made of bile salts, phospholipids, and cholesterol (Ch). They are among the most important biological entities in mammals, exhibiting an outstanding capability for solubilizing lipophilic molecules [21,22]. The study of nanoparticles for drug delivery applications allows both the development of novel platforms for the efficient transport and the controlled release of drug molecules in the harsh microenvironment (changes of pH and/or temperature) of diseased tissues in living systems [23]. Thus, these systems are being exploited for the development of pharmaceutical formulations with improved delivery to the target [24,25,26]. Lipophilicity and membrane partition properties of potential drug candidates are key to the recognition of their target in cells [27]. Despite the advantages of using nanosized drug delivery systems, some drawbacks can be found from this methodology. For example, delivering a sufficient amount of these external agents is required in order to provide the best possible efficacy. Moreover, since these nanocarriers could be affected by changes in pH or temperature in different organs, tissues, and subcellular compartments, undesired degradation of these agents could be produced with the corresponding toxic exposure, including swelling of surrounding tissues [23,28].

In previous years, the development of strategies focusing on photon Up-Conversion (UC) processes for photothermal therapy [29,30], cell imaging [31,32] and pH-responsive [33,34] treatments have received great attention [35]. These UC processes are based on the absorption of photons leading to the emission of another photon whose energy is higher than the energy of the absorbed photons. The system involves a light absorber named sensitizer or donor (such as Yb) and other species which collects the absorbed energy from the sensitizer, named acceptor (such as Er/Tm/Eu/Ho) embedded on a transparent matrix (such as NaYF_4_, BaYF_5_, NaGdF_4_, or LiYF_5_) [36]. In a UC system, initial irradiation with photons of low energy results in the emission of higher-energetic photons [37,38,39]. These materials can be tuned by modifying their photophysical properties and are able to convert intense light from the near-infrared region to high energy light in the visible or UV region [40,41,42]. It is important to note the strong stability of these UC particles after receiving NIR radiation, which allows for combining this process with others. For all these reasons, they can be used as drug delivery systems, upon light absorption, which requires the controlled release of active compounds at the specific targets responsible for the pharmacological action [43,44].

A key aspect that directly affects both the physical-chemical and photoluminescence properties of the resulting UC is the synthesis method. State of the art processes mainly consider one of the following options: coprecipitation [45], sol-gel process [46], thermal decomposition [45], or solvothermal process [46]. Coprecipitation and the sol-gel process are quite simple methods but do not allow control of the obtained crystal phase ratios, which is one of the aims to optimize the luminescence of the material. Thermal decomposition produces high-quality UCs but requires high temperatures and contaminant solvents, which are not suitable for biomedical applications. Finally, the solvothermal process is the most adequate for this study because it allows for the fine control of the structural, morphological, and optical properties of materials [47,48,49]. In addition, unlike other methods, it does not use volatile solvents that require critical synthesis temperatures and pressures conditions which increase costs, as well as decrease the sustainability of the process.

Ciprofloxacin (CF) is an antibiotic that belongs to the quinolone family and has been widely employed in the last thirty years to treat a number of bacterial infections [50] as well as prevent the formation of prostate or lung cancer cells by the inhibition of topoisomerase II [51,52]. Moreover, it is also commonly used for the elimination or reduction of bacteria from the tumor environment that consumes gemcitabine, a very common drug in chemotherapeutic treatments for breast, ovarian, or metastatic cancer [53,54]. The CF is slightly water-soluble and it can occasionally be detected in urine and blood. To achieve a specific delivery of this drug to the target, the formation of a prodrug with better water solubilization is warranted which may release the corresponding drug after an external input such as light; however, the prodrug only absorbs UV-blue light which is a biologically incompatible wavelength.

To overcome this drawback, an interesting strategy is the use of UC materials that can be excited with near-infrared (biologically compatible) light, leading to specific UV-blue emissions which would induce a specific bond cleavage of the prodrug and, therefore, efficient delivery of the active drug (CF). To demonstrate this hypothesis, we effectively illustrate the use of stable NaYF_4_:Yb,Tm as a photon UC material with optimized blue luminescence that efficiently induces CF delivery from its prodrug (an oxime-ester derivative, CF-OE) in acid media (Figure 1). The optimization of the blue emission of NaYF_4_:Yb,Tm upconverter materials has been achieved by the modification of synthesis conditions (reaction time, stabilization temperature, and pH). Obtained systems were characterized by photoluminescence (PL), X-ray diffraction (XRD), and scanning and transmission electron microscopy (SEM and TEM). In addition, to determine their photophysical behavior, the best sample was characterized by time-resolved fluorescence and transient absorption spectroscopy (TAS).

## 2. Materials and Methods

### 2.1. Synthesis of Np-UCs

All NaYF_4_:Yb/Tm samples were prepared by solvothermal synthesis. Ln(NO_3_)_3_ was used as stock solution as it contains the trivalent ions of lanthanides for the designed UC. First, the corresponding metal oxide (Ln_2_O_3_) is dissolved in nitric acid at 40−50 °C while undergoing vigorous stirring. The resultant solution is mixed at room temperature for 15 min with oleic acid and ethanol, used as growth controller and solvent, respectively. Next, fluorine and sodium (NH_4_F, NaOH) sources are added to the mixture drop by drop while stirring and the resulting solution stirred for 30 minutes. Next, the solution is introduced into an autoclave (reaction tank) into a conventional oven and then submitted to the solvothermal process at 230 °C, in order to ensure a minimum of 190 °C in the reaction tank. Finally, the UCs are washed multiple times using a mixture of ethanol/water in a centrifuge at 6000 rpm. Occasionally, additional thermal treatment where the UCs are calcined at 400 °C for 4 h is performed.

### 2.2. Synthesis of 7-(4-(tert-butoxycarbonyl)piperazin-1-yl)-1-cyclopropyl-6-fluoro-4-oxo-1,4-dihydroquinoline-3-carboxylic Acid (CF)

A modification of a reported method was used [55]. Ciprofloxacin (2 g, 6.0 mmol) and di-tert-butyl dicarbonate (1.4 g, 6.6 mmol) was dissolved in THF (60 mL). Next, a solution of sodium hydroxide (0.48 g, 12.0 mmol) in water (12 mL) was added, and the mixture stirred overnight at room temperature. After that time, the solvent was removed in vacuo, and the residue was taken into aqueous saturated ammonium chloride (200 mL). The aqueous phase was extracted with DCM (200 mL) three times. The organic phase was washed with brine (400 mL), dried over anhydrous MgSO_4_, and filtered (see Appendix A). The solvent was evaporated in vacuo to obtain 2.51 g (97%) of white solid. ^1^H-NMR (300 MHz, CDCl_3_): δ 14.94 (s, 1H), 8.71 (s, 1H), 7.96 (d, J = 12.9 Hz, 1H), 7.35 (d, J = 7.1 Hz, 1H), 3.72–3.61 (m, 4H), 3.59–3.50 (m, 1H), 3.34–3.21 (m, 4H), 1.49 (s, 9H), 1.41–1.35 (m, 2H), and 1.26–1.15 (m, 2H).

*Benzophenone oxime:* Sodium acetate (0.50 g, 6.1 mmol) was dissolved in water (4 mL) and added over a solution of hydroxylamine hydrochloride (0.42 g, 6.1 mmol) in water (4 mL). The resulting mixture was added over benzophenone (1 g, 5.5 mmol) dissolved in ethanol (10 mL), and heated at 35 °C. Next, the sample was stirred for 8 h at 80 °C. After that time, the reaction was cooled at room temperature, and cold water was added to allow the precipitation of 1.04 g (96%) of the oxime as a white solid. ^1^H-NMR (300 MHz, CDCl_3_): δ 7.5–7.43 (m, 7H) and 7.37–7.30 (m, 3H). ^1^H-NMR spectrum in agreement with published data [56].

### 2.3. Tert-butyl 4-(1-cyclopropyl-3-((((diphenylmethylene)amino)oxy)carbonyl)-6-fluoro-4-oxo-1,4-dihydroquinolin-7-yl)piperazine-1-carboxylate (CF-OE)

Tert-butyl 4-(1-cyclopropyl-3-((((diphenylmethylene)amino)oxy)carbonyl)-6-fluoro-4-oxo-1,4-dihydroquinolin-7-yl)piperazine-1-carboxylate (CF-OE): To a solution of compound 1 (0.2 g, 0.46 mmol) in dry DCM (5 mL) under argon atmosphere, oxalyl chloride (0.17 g, 1.38 mmol) and one drop of DMF were added. The reaction was stirred for 2 h at 40 °C. After that time, the solvent was removed in vacuo to give 0.21 g (100%) of an orange solid that was immediately used in the next step. Benzophenone oxime (0.09 g, 0.46 mmol) and pyridine (74 μL, 0.92 mmol) were added to a solution of the acyl chloride prepared in the previous step (0.21 g, 0.46 mmol) in dry DCM (5 mL) under argon atmosphere. The reaction was stirred at room temperature for 4 h (see Appendix A). Next, the solvent was removed in vacuum, and the resulting residue was purified by column chromatography (Hexane:THF 1:1), to give 0.17 g (59%) of white solid. ^1^H-NMR (400 MHz, CDCl_3_): δ 7.96 (s,1H), 7.83 (d, J = 13.2 Hz, 1H), 7.62–7.58 (m, 2H), 7.53–7.45 (m, 5H), 7.44–7.39 (m, 1H), 7.37–7.31 (m, 2H), 7.16 (d, J = 7.1 Hz, 1H), 3.64–3.57 (m, 4H), 3.37–3.30 (m, 1H), 3.20–3.14 (m, 4H), 1.49 (s, 9H), 1.20–1.13 (m, 2H), and 0.94–0.87 (m, 2H). ^13^C-NMR (100 MHz, CDCl_3_): δ 172.6, 164.5, 161.3, 154.7, 151.7, 147.7, 144.3, 137.8, 135.0, 133.2, 130.8, 129.6, 129.2, 129.1, 128.4, 128.4, 122.8, 113.0, 108.2, 105.3, 80.2, 50.0, 43.6, 34.7, 28.5, and 8.1 (Appendix A). HR-MS ([M + H]+): Calcd. for C_35_H_35_FN_4_O_5_ + H: 611.2664; Found: 611.2659 (Appendix A)

### 2.4. Characterization Methods

All prepared UCs were characterized by powder X-ray Diffraction (XRD) by Philips PW 3040/00 X`pert MPD/MRD, using CuKα radiation (λ = 1:54,178 Å) at a scanning rate of f 0.2/s. SEM micrographs were obtained using a field emission scanning electron microscope (Hitachi TM-1000) and a high-resolution scanning electron microscope (Nova Nano SEM230). Transmission electron microscopy (Philips Tecnai 20 T) to characterize the absorption spectra of UCs a Perkin Elmer Lambda 1050 UV equipment was used. To characterize the emission spectra of the UCs, the compound was irradiated with a laser of 980 nm (Power 1 kW, model LD-WL206, input 85–264 V, 47–63 Hz 0.4 A) and its response was measured with a Fluorescence Spectrometer Perkin Elmer LS 55.

### 2.5. Photophysical Measurements

The solvents HPLC grades were purchased from Scharlab. Milli-Q^®^ water was used for sample preparation. The stock solutions of CF and CF-OE were adjusted to 300 μg/mL in CH_2_Cl_2_. In the case of CF-OE, solutions containing 10% (*v/v*) of formic acid (FA) were added to protonate all the oxime. At this concentration of FA, the absorbance spectrum CF-OE suffers a hypsochromic shift.

UV-visible spectra for CF or CF-OE liquid samples were registered at room temperature using 10 × 10 mm^2^ quartz cells with a spectrophotometer model Perkin Elmer Lambda 1050 UV/Vis/NIR. Steady-state fluorescence experiments for CF or CF-OE liquid samples were carried out with a Fluorescence Spectrometer Perkin Elmer LS 55, with an excitation wavelength of 330 nm and using a cut-off filter at 350 nm. Time-resolved (TR) fluorescence for CF or CF-OE liquid measurements were recorded in a Mini Tau system provided with a bandpass filter of 450 nm using an EPL-375 picosecond pulsed diode laser with emission at 372 nm as excitation source with a laser pulse width of 61 ps (both from Edinburgh Instruments, Livingston, UK). For monitoring the TR fluorescence of the solid powdered Np-UC, the powder sample was irradiated with a laser of 980 nm (Power 1 kW) in front-face mode, detecting the kinetic traces with a bandpass filter of 450 or 650 nm.

Transient Absorption Spectroscopy (TAS), i.e., laser flash photolysis measurements, were carried out with LP980 equipment from Edinburgh Instruments (LP980), based on an optical parametric oscillator (OPO) pumped by the third harmonic of an Nd:YAG laser (EKSPLA, Vilnius, Lithuania). The selected excitation wavelength for the measurements was 355 nm with single low energy pulses of 1 mJ/pulse of ca. 5 ns duration, while a pulsed xenon flash lamp (150 W) was employed as a detecting light source. The probe light is dispersed through a monochromator (TMS302-A, grating 150 lines/mm) after it has passed the sample and then reaches a PMT detector (Hamamatsu Photonics, Hamamatsu Japan) to obtain the temporal profile. The absorbance of the CF of CF-OE liquid samples was kept at ∼0.3 at λ_exc_ = 355 nm in dichloromethane. All transient spectra were recorded at room temperature using 10 × 10 mm^2^ quartz cells, which were bubbled for 15 min with N_2_ before acquisition. TAS signals for Np-UCF were monitored at λ_exc_ = 980 nm in the optical parametric oscillator mode in purged organic dispersed solutions.

### 2.6. Light-Induced Drug Delivered Experiments

A CH_2_Cl_2_ solution of CF-OE (0.5 mM, 300 μg/mL) containing 10% of FA was prepared. The pH of the solution was ca. 4. Simultaneously, 10 mg of NaYF_4_:Yb,Tm (UC3) was deposed in an HPLC vial and 1 mL of the CF-OE solution was added. The resulted suspension was purged under N_2_ for 15 min. Then, the solution was irradiated (λ_exc_ = 980 nm) at a distance of 5 cm for increasing periods (0–100 min). The solution was maintained by continuous stirring at room temperature. Then, aliquots of 30 μL of the suspended solution were diluted until 1 mL in acetonitrile and filtered before HPLC analysis. The amount of CF and CF-OE, as well as the potential release of the corresponding free CF upon the irradiation reaction, were determined using a reversed-phase HPLC Jasco LC-4000 series system, equipped with a PDA detector MD-4015 and a multisampler AS-4150 controlled by ChromNav software (Jasco Inc., Ashikaga, Japan). A Purple ODS reverse-phase column (5 µm, 4.6 × 150 mm, Análisis Vínicos SL, Ciudad Real, Spain) was employed. For the quantification of all the chemical species, isocratic conditions were used. The flow rate was 1 mL/min, and the column temperature was fixed at 25 °C. In all cases, the injection volume was 10 µL. The mobile phase was based on a mixture of 70:50 H_2_O:CH_3_CN, containing 0.1% of FA. The retention times obtained from the analysis of CF or CF-OE were 4.9 and 10 min, respectively. The calibration curve was reached at the concentration of solutions standard of CF or CF-OE in the range of 0.5–30 μg/mL. In parallel, the reaction was monitored in-situ by TAS, in order to check the presence of free CF and Benzophenone as products formed.

## 3. Results and Discussion

### 3.1. Optimization of the Synthesis of NaYF_4_:Yb/Tm UC Materials

In the first stage, optimization of the synthesis conditions such as reaction time, thermal treatment, and pH of the reaction (Table 1) was performed to determine optimal parameters to prepare NaYF_4_:Yb/Tm UC materials with an improved emission on the blue-light range of the spectrum. These modifications lead to changes in structural and morphology properties that have an intrinsic relationship with luminescence properties.

Modifications in the synthesis time (UC1 & UC2) lead to minor differences in the visible luminescence spectra (Figure 1A). All samples show three main contributions at 480, 652, and 700 nm previously assigned in the literature to ^1^G^4^ → ^3^H^6^, ^1^G^4^ → ^3^H^4^, and ^3^F^3^ → ^3^H^6^ transitions, respectively (see Figure 1A inset) [57], where the IR contribution show a decrease for short reaction times (UC1). On the other hand, the absence of thermal treatment (UC3) leads to a dramatic effect in emission spectrum (Figure 1A) with a huge increase of six times in the luminesce intensity for the 480 nm emission and the presence of two new bands at 360 nm and 456 nm attributed to ^1^D^2^ → ^3^H^6^ and ^1^D^2^ → ^3^H^4^ transitions, respectively (Figure 1A inset).

Regarding structural properties, lower reaction times (UC1) lead to a mixture of two crystal phases known as α or cubic-phase (43.1%) and β or hexagonal-phase (56.9%). Conversely, higher reaction times (UC2 & UC3) exhibit the formation of hexagonal one as a single phase (Figure 1B). This behavior is in concordance with previous studies that show that the β-phase is energetically more favorable [58]. Thus, for a shorter reaction time, the synthesis is kinetically controlled while a thermodynamic control is most favorable for a higher one [59]. Similar results were obtained in absence of thermal treatment also corroborating this fact. On the other hand, morphological properties studied by SEM (Figure 1C) and TEM images (Appendix A), exhibit that in the UC1 sample, two well-differentiated morphologies are observed: faceted crystals with the hexagonal habit (1.5–3 µm) corresponding to the β-phase and smaller irregular crystals with a globular shape (0.2–0.3 µm) assigned to the α-phase. An increase in the particle size of the β-phase (2–3.5 µm) is detected for a longer synthesis time (Figure 1B). In addition, the absence of thermal treatment leads to a decrease in the particle size of (0.8–1.2 µm) the β-phase which maintains the globular shape. These results indicate that the presence of a hexagonal phase with smaller particle size gives rise to an improvement in the luminescence yields in blue emissions and favors emission at lower wavelengths [60].

The pH modulation from 5 to 10 (UC3–UC7) leads to critical changes in the UC properties. Thus, for a pH higher than 6, the luminescence intensity of all emissions dramatically decrease, being negligible for pH ≥ 7 (Figure 2A). These changes in the synthesis media lead also to modifications in the structural and morphological properties. Although, XRD patterns show the presence of a β-phase in all cases (Figure 2B), SEM and TEM studies (Figure 2C and Appendix A) show significant changes in the shape and size of UC materials with the pH increase. Thus, in the case of a basic environment (UC6–UC7), the materials show well-defined prismatic hexagonal single crystals with sizes between 1–2 µm long and 0.2–0.5 µm wide. On the other hand, for acidic and neutral media (UC3–UC5), the particles exhibit a globular shape with smaller sizes between 1–1.5 µm long and 0.1–0.3 µm wide. These changes in morphologic properties are observed with changes in the pH and are also attributed to the presence of NaOH. Previous studies proposed that Na+ ions react with the oleic acid-forming oleates, which are miscible with the reaction medium favoring the precipitation of particle seeds in preferred orientation forming well-defined hexagonal crystals [61,62].

Therefore, changes in luminescence could be attributed to both structural and morphological properties. Thus, higher crystal sizes and crystallinity lead to a decrease in the emission properties of NaYF_4_:Yb/Tm [63].

### 3.2. Light-Driven Drug Delivery

The performance of NaYF_4_: Yb as photoactive materials for drug delivered therapies in Ciprofloxacin photosensitization reactions was explored. As abovementioned, the development of a better water-soluble derivative of CF is needed in order to successfully perform its delivery. In addition, this candidate should contain an adequate leaving group that could be easily removed after absorbing light. Moreover, in drug delivery experiments, it is crucial that the drug is efficiently released in the target to allow for drug absorption where its pharmacological effect is required. The absorption of CF progressively decreases from the stomach (100%) to the descending colon (5%) [64,65].

To fulfill these requirements, the oxime ester CF-OE (see chemical structure in Figure 1) was chosen as a model compound that was obtained following the corresponding synthetic strategy (see details in Appendix A). This prodrug CF-OE was found to be soluble in water with a maximum absorption band at around 425 nm observed in acidic media due to its protonation (Figure 3A and Appendix A). Thus, activation of this molecule by blue emission would afford the drug CF and the leaving group through the N–O bond cleavage. Interestingly, the emission spectrum of NaYF_4_:Yb,Tm (UC3) showed a clear superposed signal with the protonated CF-OE absorption spectrum in the broad range of 460–500 nm (Figure 3A). This indicated that UC3 was a suitable system for photoinduced drug delivery.

To corroborate the suitability of this reaction, we also examined its photoluminescence properties. While CF exhibited a high fluorescence with the maximum at ca. 450 nm, the corresponding oxime-ester resulted in a poor intensity, accompanied by a blue-shift of the maximum (Figure 3B). For the protonated CF-OE (10% FA), the fluorescence was still lower, accompanied by a red-shift of the maximum up to 460 nm, as reported in the literature for commercial ciprofloxacin [66]. On the other hand, time-resolved emission (Figure 3B inset) shows identical kinetic traces (τ_F_ = 575 ps) for both CF and CF-OE (including in acidic media). The transient absorption spectrum of CF (Appendix A) showed the main contribution at 430 nm due to the fluorescence signal that avoids the accurate determination of the transient for CF. For this reason, TAS experiments for CF, CF-OE, and CF-OE in acidic media were monitored 100 ns after laser pulse (Appendix A). These transients showed an identical absorption in the overall spectrum window, with a slight growth in the signal from 600 nm. The kinetic trace (λ_mon_ = 650 nm) showed a large lifetime (5.8 μs) which could be assigned with a radical of CF formed after laser pulse (Appendix A inset).

The controlled drug release process through photosensitized N-O bond cleavage in CF-OE was investigated upon selective irradiation of Np-UCs (UC3, λ_exc_ = 980 nm) in the presence of CF-OE in organic acidic media (Figure 4). First, to ensure accurate HPLC monitoring, the starting CF-OE and the corresponding free CF were tested and the calibration curve was reached at the concentration of solutions standard of CF or CF-OE in the range of 0.5–30 μg/mL (Appendix A). In the light-driven drug delivery reaction, a remarkable increase in the formation of CF was observed at the first stages of the reaction accompanied by the corresponding decrease in the CF-OE chromatogram signal reaching a practically quantitative conversion (Figure 4A and Appendix A). This behavior could be attributed to the N–O bond splitting, affording the free CF and the imine compound that immediately hydrolyzes to give the corresponding benzophenone (Appendix A). As expected, no changes were observed in absence of Np-UCs at optimal conditions.

Furthermore, the stability of the UC3 sample was also investigated by reusability experiments. After each experiment, the supernatant of the mixture solution was removed, and the recovered UC NPs were washed with CH_2_Cl_2_ several times. After that, an aliquot of the last supernatant was analyzed by UV-vis spectroscopy to check the presence of the drugs. Figure 4B and Appendix A show the UC3 activity during five consecutive cycles, with only a 3% loss of efficiency.

In addition, to gain further insight into the reaction mechanism, in-situ TAS experiments were performed in the presence of UC3 and released CF under continuous 980 nm irradiation. To monitor the performance of UC3 and drugs, TAS was conducted using two excitation wavelengths. At λ_exc_ = 980 nm, a transient band at ca. 356 nm was detected (Figure 5A) which was safely ascribed to fundamental absorption from ^3^H^6^ to ^1^D^2^ excited states of Tm^3+^ [67]. This transient completely disappears upon 10 ns after the laser pulse and appears again when the experiments are repeated. On the other hand, at λ_exc_ = 355 nm, the starting oxime and the released products were monitored. Since the identical TA signal was registered for free CF and CF-OE (Appendix A), the reaction was followed by the formation of benzophenone (BP). The photolysis of BP generates the corresponding triplet excited state (^3^BP*) with a characteristic band centered at 520 nm which increased with the irradiation times (Figure 5B).

## 4. Conclusions

Here, we have demonstrated that NaYF_4_:Yb,Tm up-converted material is an excellent candidate for blue light-driven drug release under near-IR wavelength. The emission properties of these materials were improved through an optimization of the synthesis procedure (reaction time, thermal treatment, and pH conditions). The best blue-emission performance was achieved for the UC3 sample prepared through 24 h-synthesis without thermal treatment at a pH of 5. The enhancement of the blue-emission is attributed to the presence of the β-phase and smaller particle size.

Drug delivery experiments were performed through the highly effective photosensitization, by the means of NaYF_4_:Yb,Tm up-converted material, of the N-O bond cleavage of the oxime ester of Ciprofloxacin (prodrug). Stability studies show that UC3 is a very stable and easily recovered material and can be used in several reaction cycles. This methodology can be extended to other drugs containing photoactive chromophores and is available as an alternative for drug delivery systems that require a controlled release of the active compounds at specific targets responsible for pharmacological action.

## Data Availability

The data presented in this study are available in the article.

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
