# Peer review of "Controlled Synthesis of Up-Conversion NaYF4:Yb,Tm Nanoparticles for Drug Release under Near IR-Light Therapy"

_biomedicines, 2021, doi:10.3390/biomedicines9121953_

Round 1

Reviewer 1 Report

In the present manuscript the authors reported a controllable synthesis of NaYF4:Yb,Tm up-converted (UC) material to improve its photoluminescence properties. This system was then studied as drug delivery system to achieve a drug delivery system for ciprofloxacin. In my opinion the work is well performed and new and therefore it deserves to be published on Biomedicine after revision. 1. First of all some part of the Results and Discussion should be improved to improve the overall understandability. In particular the section 3.2. I suggest the authors to better introduce the modification of ciprofloxacin for their purpose. 2. Introduction part should be shortened, I suggest the authors to report only what if strictly necessary for their objectives. 3. Have they studied the cytotoxic effects of the NaYF4:Yb systems? 4. How they suppose to be the delivery of CF by means of the UC systems obtained? 5. Pag 5 and Figure S10 are not present in the provided version. 6. In paragraph 2.1 what the authors means with flour?

Author Response

Answer to Referees
Referee 1:

In the present manuscript the authors reported a controllable synthesis of

NaYF4:Yb,Tm up
-converted (UC) material to improve its
photoluminescence properties. This system was then studied as drug

delivery system to achieve a
drug delivery system for ciprofloxacin. In my
opinion the work is well performed and new and therefore it deserves to be

published on Biomedicine after revision.

We would like to thanks to the referee for the constructive comments.

1.
First of all some part of the Results and Discussion should be improved
to improve the overall understandability. In particular the section 3.2. I

suggest the authors to better introduce the modification of ciprofloxacin for

their purpose.

In other to clarify the section 3.2 here you can find a more extensive explanation about
the necessity to include the oxime derivative (CF-OE).

It is knowing that the absorption of ciprofloxacin (CF) is not the same in all regions of the
human gastrointestinal tract. Thus, while the release in the stomach is high, the
absorption of the drug is progressively decreasing in the jejunum (ca. 37%), ileum (23%),
the ascending colon (7%) and the descending colon (5%). (Ciprofloxacin absorption in
different regions of the human gastrointestinal tract. Investigations with the hf-capsule,

https://doi.org/10.1111/j.1365-2125.1990.tb03740.x
and Absorption differences of
ciprofloxacin along the human gastrointestinal tract determined using a remote-control
drug delivery device (HF-capsule),
https://doi.org/10.1016/0002-9343(89)90026-0).
This is the reason because the present work is a success proof of concept of the drug
delivery of CF from the corresponding prodrug (CF-OE) through N-O photosensitized
process. This studies can be extended in future works to the experiments under
physiologic condition.

On the other hand, even when direct visible light could be also used to activate the
prodrug, the use of UC system allows to irradiate the sample with infrared wavelengths
which are ideal to work in the first and second biological Windows (650-1350 nm).

Finally, the prodrug is inactive because the acid group is blocked, therefore the
administration of CF-OE allow to maintain the drug activity blocked and then activated
by the illumination.

In addition, we have included an introductory text included in section 3.2

To explore the performance of NaYF4: Yb as photoactive materials for drug delivered
therapies in Ciprofloxacin photosensitization reactions. As abovementioned, the devel-
opment of a better water-soluble derivative of CF is needed in order to successfully per-
form its delivery. In addition, this candidate should contain an adequate leaving group
which could be easily removed after absorbing light. Moreover, in drug delivery experi-
ments is crucial that the drug must be efficiently released in the target allowing the drug
absorption where its pharmacological effect is required. Since the absorption of CF pro-
gressively decrease from stomach (100%) to descending colon (5%) [61,62].”

To fulfill these requirements, the oxime ester CF-OE (see chemical structure in Scheme
1) was chosen as model compound that was obtained following the corresponding syn-
thetic strategy (see details in Scheme S1). This prodrug CF-OE was found to be soluble
in water and a maximum absorption band at around 425 nm was observed in acidic
media due to its protonation (Figure 3A and Figure S5). Thus, activation of this molecule
by blue emission would afford the drug CF and the leaving group through the NO bond
cleav-age. Interestingly, the emission spectrum of NaYF4: Yb, Tm (UC3) showed a clear
super-posed signal with the protonated CF-OE absorption spectrum in the broad range
of 460-500 nm (Figure 3A). This evidently indicated that UC3 was the suitable system
for photoinducing the drug delivery.

2.
Introduction part should be shortened, I suggest the authors to report
only what if strictly necessary for their objectives.

Following the reviewer comment, we have modified the introduction (highlighted

in yellow in main text).

3. Have they studied the cytotoxic effects of the NaYF4:Yb systems?

Previous
studies have shown that Up Converter (UC) materials, in general, are
not considered to be highly toxic materials, but the release of fluorides and

lanthanides upon their dissolution may cause cytotoxicity
[1,2]. Their
disintegration is particularly worrying in highly diluted dispersions of nanoparticles

because both the lanthanide ions and/or the bare UCNPs can cause undesirable

interference in a chemical or biological environment.
To avoid this potential
proble
ms, several groups have been developed coating strategies using bio-
compatible polymers [3,4].
Additional advantages of the use of PSS as capping
layer are its biocompatibility and its high dispersibility in water, together with easy

f
urther functionalization of this kind of nanohybrids.
The objective of our work is to demonstrate that these systems are highly

effectives for drug delivery. In further studies we planned to prepare UC

nanoparticles, which will be
coated with biocompatible polymers such as PSS
and
functionalized with CF to select a nanohybrid with optimal size, drug loading
and stability. Citotoxicity and cell viability studies would be also performed as

previously reported for other upconversion nanohybrids.

Bibliography

1.
Feng Duan, Y. C. (2012). Recent advances in synthesis and surface
modification of lathanide
-doped upconversion nanoparticles for biomedical
applications.
Biotechnology Advances (2012) 30 (6), 1551-1561
2.
J. Chen, J. X. (2012). Upconversion nanomaterials: synthesis, mechanims,
and applications in sensing. Sensors, 2414
-2435. doi: 10.3390/s120302414
3.
Nestor Estebanez, María Gonzalez-Bejar,and Julia Perez-Prieto;
Polysulfonate Cappings on Upconversion Nanoparticles Prevent
Their
Disintegration in Water and Provide Sup
erior Stability in a Highly Acidic Medium,
ACS Omega 2019, 4, 3012−3019

4. Juan Ferrera-González; Laura Francés-Soriano, Cristina Galiana-Roselló,
Jorge González
-Garcia, María González-Béjar, Eleonore Fröhlich, and Julia
Pérez
-Prieto, Initial Biological Assessment of Upconversion Nanohybrids,
Biomedicines 2021, 9,
1419.
4. How they suppose to be the delivery of CF by means of the UC systems

obtained?

This is an interesting question.
After UC system implementation it can be
irradiated by means several innovative
fiber optic illumination technologies, such
as i
ntraoperative guided surgery (IGS), where the use of near-infrared (NIR)
fluorescence guides
is one of the latest trends [1-3], for use in both fundamental
medical research and clinical practice
, due to advantages in reduction of light
scattering, photon absorption and
autofluorescence via broadening. In addition,
due to their characteristics UC systems are
ideal to work in the first and second
biological Windows
(650-1350 nm).
1.
Zhibei Qu, Jianlei Shen, Qian Li, Feng Xu, Fei Wang, Xueli Zhang,
Chunhai Fan; Near
-IR emissive rare-earth nanoparticles for guided
surgery
, Theranostics 2020, 10 (6) 2631
2.
Jiayu Wang and Jianfei Dong1, Optical Waveguides and Integrated
Optical Devices for Medical Diagnosis, Health Monitoring and Light

Therapies
Sensors (Basel). 2020 20(14): 3981.
3.
Andrew M. Smith, Michael C. Mancini & Shuming NieSecond window for
in vivo imaging. Natur
e. (2009) 710-711. 10.1038/nnano.2009.326
5. Pag 5 and Figure S10 are not present in the provided version.

Sorry for this mistake, may be Pag 5. was erased in the pdf production process.

Regarding to Figure S10 is cited on Page 9 Line 356

6. In paragraph
2.1 what the authors means with flour?
Many thanks to highlight this error. This is a grammatical
mistake. Fluor means
Fluorine.

Author Response

Answer to Referees
Referee 2:

Review report on the manuscript titled Controlled Synthesis of Up
-
Conversion NaYF4:Yb,Tm Nanoparticles for Drug
Release Under Near IR-
Light Therapy. In this paper, controllable synthesis of NaYF4:Yb,Tm up
-
converted (UC) material to improve its photoluminescence properties has

been performed. NaYF4:Yb,Tm up
-converted material promotes of the N-O
bond cleavage of the
oxime ester of Ciprofloxacin. HPLC chromatography
and transient absorption spectroscopy measurements were performed to

evaluate the drug release conversion rate. Although the manuscript

emphasize interesting fabrication approach based on solvothermal

metho
ds, there are some concerns related to the suitability of these methos
for drug delivery purpose.

We would like also thanks to referee 2 his/her positive comments.

1.
The introduction section is too large. It should be better to resume the
general part and to
present concise information on the advantages of
solvothermal methods for preparation of NaYF4:Yb, Tm nanoparticles for

Ciprofloxacin release. Maybe a comparison with other delivery methods

would be useful.

We have modified the Introduction to focus in th
e most important issue.
Regarding to the comparison with other drug delivery methods, this information

is collected in the introduction.

The release of drugs through the blood to the target is a field widely investigated
in last decades [1,2]. The use of drug delivery systems by means of liposomes

[3,4], mixed micelles [5
7], niosomes [8], micelles [9], bile salts aggregates [10
12], nanopa
rticles [13,14], nanocapsules, gold nanoparticles, microspheres [15],
microcapsules, nanobubbles [16], microbubbles and dendrimers [17,18] is being

investigated for diagnosis and therapy [19,20]. Carriers based on micelles

(liposomes, mixed micelles, nioso
mes, among others) are made of bile salts,
phospholipids and cholesterol (Ch). They are among the most im
-portant
biological entities in mammals, exhibiting, for instance, an outstanding capability

for solubilizing lipophilic molecules [21,22]. The study o
f nanoparticles for drug
delivery applications allows both the development of novel platforms for the

efficient transport and the controlled release of drug molecules in the harsh

microenvironment (changes of pH, temperature) of diseased tissues of living

systems [23]. Thus, these systems are being exploited for the development of

pharmaceutical formulations, with improved delivery into the target [24
26].
Lipophilicity and membrane partition properties of potential drug candidates are

key to the recognitio
n of their Target in cells [27]. Despite advantages of using
nanosized drug delivery systems, some drawbacks can be found from this

method
-ology. For example, delivering a sufficient amount of these external
agents is highly re
-quired in order to provide the best possible efficacy. Moreover,
since these nanocarriers could be affected by changes in pH or temperature in

different organs, tissues, and sub-cellular compartments, undesired degradation
of these agents could be produced, with the corresponding tox
ic exposure,
including swelling over surrounding tissues [23,28]
.”
It is not clear what type of kinetic profile is associated to Ciprofloxacin

release?

Many thanks for the referee comment. May be the confusion is because we use

the kinetic terminology for several applications in this study.

From one side, Figures Figure 4A and S8 show the kinetic profile of

Ciprofloxacine (CF) drug release, which is moni
tored by HPLC chromatography,
and exhibit a linear behavior and quantitative conversion.

On the other hand, time resolved fluorescence and transient absorption

spectroscopy experiments were performed to determine the kinetic behavior of

excited states of t
he intermediate species during the light mediated process. In
this case, we found that UC3 materials showed a transient band at ca. 356 nm

which is attributed in the literature to fundamental absorption from
3H6 to 1D2
excited states of Tm
3+ (Figure 5A). Moreover, in situ TAS experiment, exhibit that
the photolysis of BP generates a 3BP* excited state with a characteristic band

centered at 520 nm which increased with the irradiation times (Figure 5B).

Again, is it comparable to other NPs fab
rication technique? Why this
elaborated method would be preferred instead of others?
Are the
costs/benefits advantageous?

As is reported in the introduction, one of the main aspects that affect
both the
physic
-chemical and photoluminescence properties of the resulting UC is the
synthesis method
. Several synthesis methodologies have been described in the
literature.
solvothermal was selected because is the most adequate because
offer a better control structural, morphological and optical properties of UC.
In
addition, unlike other methods, it does not use volatile solvents that require critical

synthesis temperatures and pressures
conditions, increasing costs, as well as
decreasing the sustainability of the process
.
I recommend the publication after minor
revision, taking into account the
above observations.

We hope that these corrections are enough to make the paper suitable for it

publication on Biomedicines Journal.

Round 2

Reviewer 1 Report

The authors improved the quality of their manuscript and therefore it now can be accepted for publication on Biomedicines as it is.